# Executable Visual Reasoning: From Universal Solver to Emergent Behaviors

## Abstract

Recent releases such as o3 highlight human-like "thinking with images" reasoning that combines structured tool use with stepwise verification, yet most open-source approaches still rely on text-only chains, rigid visual schemas, or single-step pipelines, limiting flexibility, interpretability, and transferability on complex tasks. We introduce **ExeVision**, which explores executable code as a universal solver for visual reasoning. Unlike fixed-schema calls (e.g., only predicting bounding-box coordinates), ExeVision defines, composes, and executes code to orchestrate multiple tools, compute intermediate results, and render visual artifacts (e.g., boxes, lines, plots) that support transparent, self-checkable reasoning. To guide this process, we introduce a reward for Balanced Adaptive Tool-call, which balances exploration with efficiency and mitigates tool overuse. Interestingly, beyond the expected capabilities taught by atomic supervision, we empirically observe novel emergent behaviors during RL training: ExeVision demonstrates novel tool invocations, unseen compositions, and cross-task transfer. These behaviors arise without task-specific fine-tuning, suggesting a general and scalable mechanism of executable visual reasoning. Extensive experiments across reasoning benchmarks (e.g., visual search, math, chart QA) show that ExeVision not only consistently outperforms schema-driven and text-only baselines, but also surpasses advanced closed models such as GPT-4o and larger open-source models. code is available.

## 1 Introduction

Multimodal Large Language Models (MLLMs) have made rapid progress, showing strong capabilities in both visual perception and reasoning. By leveraging the language-centric chain-of-thought (CoT) mechanism (Brown et al., 2020; Wei et al., 2022), models can decompose complex problems into intermediate steps, thereby improving performance on challenging tasks. However, the CoT paradigm's reliance on static context becomes a critical limitation when extended to modalities such as vision. This prevents models from interacting with visual inputs or incorporating new observations during intermediate reasoning (Zou et al., 2024; Chung et al., 2025), creating an information bottleneck that hinders multi-round focusing and validation. To address this, the o3 system (OpenAI, 2025) integrates the ability to actively seek new information through multiple tool invocations, supporting iterative reasoning over visual inputs and demonstrating strong perception and analysis.

**Research gaps.** While recent models have made notable progress, fundamental gaps remain unresolved. (1) Current approaches largely extend CoT into multimodal reasoning via text-only templates, failing to incorporate new observations, refine intermediate steps, or validate its reasoning against visual evidence (Ko et al., 2025; Feng et al., 2025). (2) In addition, o3 remains a proprietary black-box system: its internal mechanisms are inaccessible, its reasoning process is less transparent, and its outputs cannot be systematically studied or reproduced; (3) Most open-source systems incorporating visual reasoning remain restricted to predefined visual workflows, or rigid and schema-based pipelines (e.g., predicting bounding box coordinates for cropping operations), which are inherently inflexible and task-specific, limiting transfer to new tools and tasks (Zheng et al., 2025; Su et al., 2025a; Zhang et al., 2025b; Su et al., 2025b). Consequently, the field still lacks an open and verifiable medium, that is general across tools and tasks, for multimodal reasoning that allows MLLMs to dynamically compose tools, produce intermediate artifacts, and self-check their outputs in a transparent and reproducible manner. Addressing this gap is crucial for achieving flexible, explainable, and transferable reasoning across complex real-world tasks.

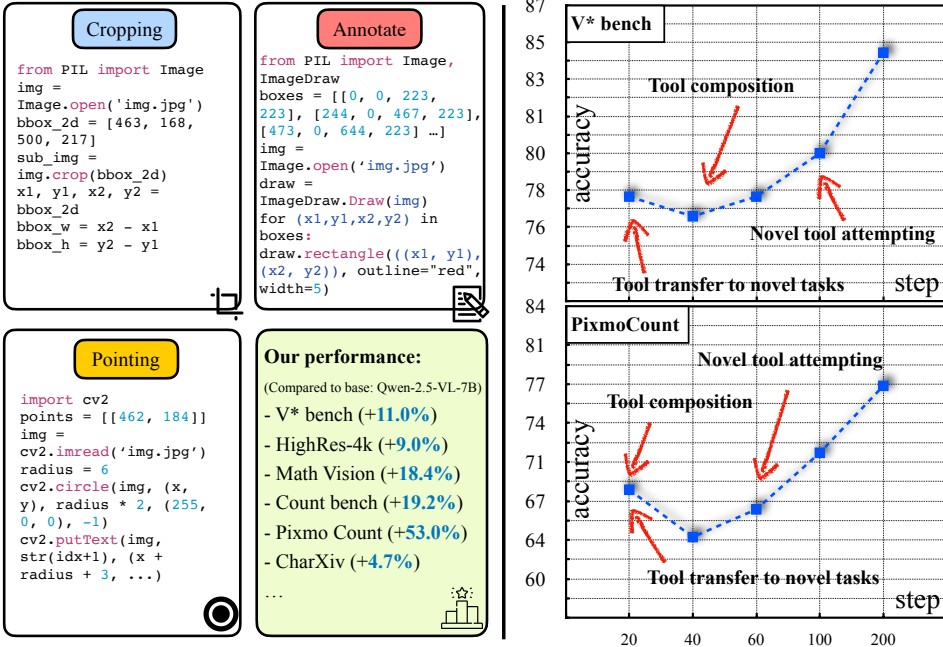

Figure 1: **Left:** In SFT stage, the model is equipped with fundamental atomic coding abilities as a universal solver (e.g., cropping, pointing, annotation) for visual reasoning, through our curated executable trajectories in sandboxed environments. **Right:** During RL, we empirically observe novel *emergent behaviors*: the model spontaneously composes previously learned operations, attempts novel tool usages, and transfers skills to unseen tasks, going beyond the capabilities covered by supervision. **Bottom:** After RL, the learnt reasoning abilities translate into consistent and significant performance gains on diverse multimodal benchmarks. Representative trajectories are in Figure 3.

In this work, we introduce ExeVision, a multimodal reasoning framework that leverages executable code as a universal medium for visual reasoning. Unlike prior schema-based pipelines with fixed operation templates, code enables the model to define, compose, and execute diverse visual–symbolic operations, producing both intermediate artifacts (e.g., cropped regions, plots, annotations) and final answers within a unified, verifiable reasoning process. To equip the model with fundamental skills, we curate a high-quality trajectory dataset and use supervised fine-tuning (SFT) to teach atomic capabilities such as counting, spatial grounding, and image annotating, enabling iterative exploration-reflection reasoning process. Building on this foundation, we employ reinforcement learning (RL) to further enhance tool-based reasoning. A central challenge we identify is a trade-off between exploration and selectivity: naïve policies often overuse tools, incurring unnecessary steps, or underuse them, failing to leverage visual interactions when needed. To address this, we design a difficulty-adaptive tool-reward mechanism that explicitly modulates incentives based on task demands, encouraging longer operation chains for genuinely complex problems while discouraging redundant calls on simpler ones. This principled reward shaping aligns the learning dynamics with the intrinsic structure of multimodal tasks, yielding a model that reasons more adaptively and transparently. Together, these components enable ExeVision to advance beyond rigid schema-based methods and offer an open, generalizable medium for executable visual reasoning.

**Empirical observation of emergent behaviors.** Although the model is only explicitly supervised on atomic operations, our design enables behaviors to emerge during RL stage that go beyond the provided supervision. In particular, we consistently observe: **(1)** Novel coding routines. The model generates procedural and computational code (e.g., clustering, function plotting) that is absent from the SFT data, indicating a capacity to internalize and extend programming patterns. **(2)** Compositional strategies. It develops coordinated routines that combine multiple atomic operations, such as localizing and counting before numerical computation, or cropping followed by rotation and annotation, that were never observed together in the training data, giving rise to higher-level strategies beyond supervised coverage. **(3)** Cross-domain transfer. The model reuses code operations learned in one supervised context to solve novel tasks where such supervision was absent, e.g., applying mathematical computation routines to answer general visual reasoning questions. Overall, these findings highlight our central insight: treating executable code as a universal reasoning medium

enables the model not only to master atomic operations but also to spontaneously develop new capabilities: tool invention, strategic composition, and transferable skills that beyond direct supervision.

**Our contributions are summarized as follows**: **(1)** We introduce ExeVision, a multimodal agent that can "think with images" by planning and composing visual–symbolic operations through executable code as a universal medium. To this end, we curate a 34K high-quality SFT dataset covering diverse atomic code capabilities (e.g., cropping, line drawing, point plotting), and additionally design a difficulty-adaptive reward mechanism for RL, enabling multi-turn reasoning and balanced tool use. **(2)** Beyond these design choices, we report novel empirical findings: despite being trained only on atomic operations, ExeVision empirically exhibits emergent behaviors during RL training, including spontaneous novel tool routines, unseen operation compositions, and cross-task transfer to novel tasks. These observations highlight the scalability and generality of executable-code reasoning beyond direct supervision. **(3)** We evaluate ExeVision on more than 10 multimodal benchmarks, spanning both general perception and complex reasoning (e.g., visual search, counting). Across the board, it outperforms advanced closed models (e.g., GPT-4o) and larger open-source baselines (e.g., Qwen2.5-VL-32B), demonstrating strong visual perception ability and broad generalizability.

## 2 RELATED WORKS

Recent progress in MLLMs aims to build systems that can effectively understand and reason across multiple modalities for complex tasks. To achieve this, current research typically focuses on three key areas: (i) enhancing perceptual capabilities across diverse modalities, (ii) improving reasoning and action capabilities through function/tool calling, and (iii) developing high-quality synthetic datasets for complex multimodal tasks where annotated trajectories are scarce. We address these aspects in our work and provide detailed discussion in the subsequent section and Appendix A.3.

**Perceptual capabilities of MLLMs.** Popular MLLMs integrate vision encoders with language models through lightweight learnable adapters, such as MLP (Zhu et al., 2023), Resampler (Alayrac et al., 2022) and Q-former (Li et al., 2023), for efficient cross-modal alignment. To enhance visual comprehension capabilities, later approaches employ visual instruction tuning combined with knowledge distillation, producing robust dense models, such as LLaVA-series (Liu et al., 2023; Li et al., 2024a) and InstructBLIP (Dai et al., 2023). Beyond dense architectures, recent state-of-the-art models continue to improve both model capacity and computational efficiency through mixture-of-experts (Shazeer et al., 2017), such as Aria (Li et al., 2024b) and Uni-MoE (Li et al., 2025). Other advancements to improve perception capabilities include accomodating high-resolution image inputs (Liu et al., 2024; Guo et al., 2024) or supporting native-resolution images (Bai et al., 2025b).

**Multimodal reasoning and tool invocation.** Building upon text-based chain-of-thought (CoT) reasoning (Wei et al., 2022; Yao et al., 2022), researchers have extended intermediate reasoning steps to multimodal settings (Zheng et al., 2025; Yeo et al., 2025) including counting (Zhang et al., 2024b), localization (Wu & Xie, 2024), charts (Li et al., 2024c), and visual math (Chen et al., 2025). To enhance reasoning capabilities, recent works integrate external tools through reasoning-and-acting frameworks (Yao et al., 2022; Yang et al., 2023), learned API usage (Schick et al., 2023), and multimodal agents that orchestrate OCR, detection, and editors (Wu et al., 2023; Shen et al., 2023). ViperGPT (Surís et al., 2023) compiles queries into executable programs. Recent models like OpenAI's o3 (OpenAI, 2025) integrate comprehensive tool capabilities directly into reasoning chains, trained via RL on large-scale CoT data. Other approaches include RL-based tool invocation (Zheng et al., 2025; Su et al., 2025a) and SFT-based methods (Wang et al., 2025). However, challenges remains such as ad-hoc operations, sparse supervision, limited task coverage and lack of comprehensive evaluation. We diverge from prior work by pursuing code as universal medium to execute multimodal reasoning across diverse atomic abilities.

## 3 METHODOLOGY

Here, we firstly introduce the fundamental preliminary in Section 3.1. An overview of our proposed ExeVision is shown in Figure 2. In Section 3.2, we further detail the reward designs made for model training and discuss their resulting benefits. In Section 3.3 and Appendix A.3, we describe the high-quality data synthesis pipeline, which covers fundamental code-based atomic operations such as cropping, counting, and math reasoning mainly tailored for SFT for initialization as cold start.

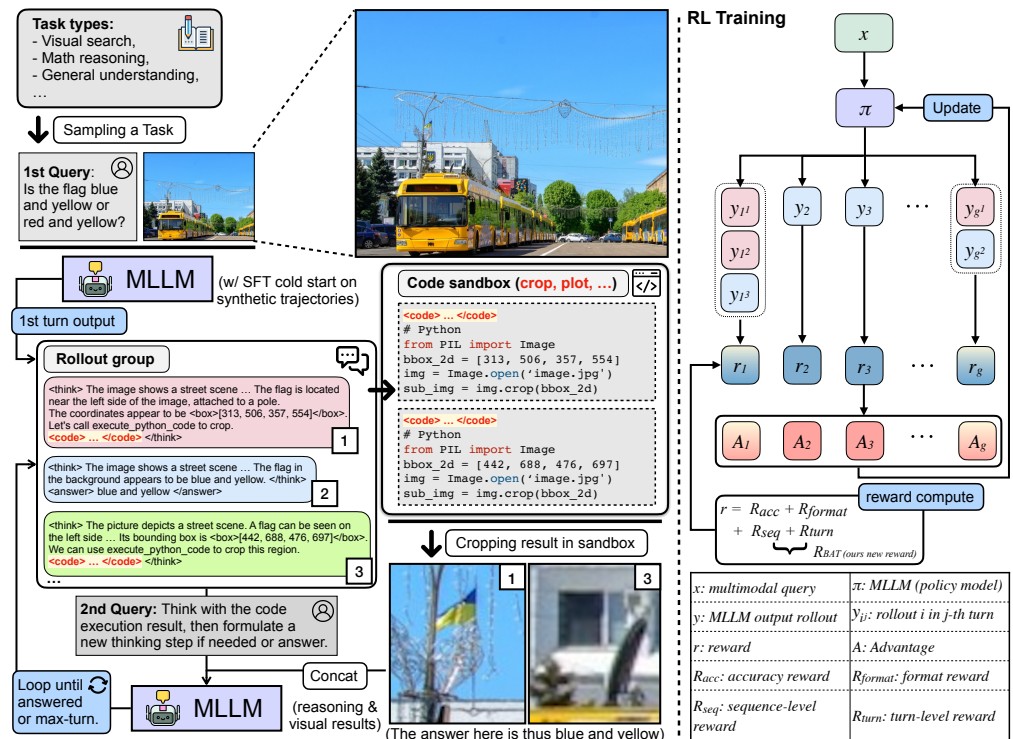

Figure 2: **Overview of our framework. Left:** A multimodal query is processed by the MLLM, which produces rollouts interleaving natural-language reasoning with executable code (e.g., cropping, plotting). Code is executed in a sandbox, and the resulting visual evidence is concatenated with text to refine reasoning or yield the final answer (e.g., "blue and yellow"). **Right:** In RL training, the policy model generates multiple rollouts that are scored by rewards for accuracy ($R_{\text{acc}}$), format compliance ($R_{\text{format}}$), and tool usage ($R_{\text{seq}}, R_{\text{turn}}$). The aggregated signal defines the advantage $A$ for policy updates, closing the loop toward verifiable executable reasoning. Details are in Section 3.

### 3.1 PRELIMINARY

**Multimodal CoT.** While chain-of-thought (CoT) reasoning improves interpretability in text-only settings, it remains static and lacks exploration when extended to multimodal scenarios. To address this, we define a *think–execute–feedback* cycle as the minimal reasoning unit under a policy model $\pi$, where each turn comprises (i) the current query and reasoning trace, (ii) a candidate action, and (iii) the resulting observation after code execution. Formally, a trajectory is (also see Figure 2, left side)

$$\tau = \big((s_{t_1}, a_{t_1}, s'_{t_1}), \ldots, (s_{t_{m-1}}, a_{t_{M-1}}, s'_{t_{M-1}}), (s_{t_M}, a_{\text{answer}})\big), \text{ where } t \text{ is time step.}$$

$s_t = (x, \nabla_t, \epsilon_t)$ contains the original query $x$, the accumulated reasoning trace $\nabla_t$, and interpreter feedback $\epsilon_t$. Actions $a_t$ are drawn from a space including tool calls (code snippets) and a terminal answer; executing code yields an observation and updates the state to $s'_t$. By iterating $a_t \sim \pi(\cdot \mid s_t)$ until a final answer is produced or a maximum turn budget $M$ is reached, each turn becomes an executable and verifiable reasoning unit. Building on this formulation, Section 3.3 detail the curation of a high-quality trajectory dataset encompassing diverse atomic abilities. This dataset provides the foundation for initializing the policy model through SFT, before advancing to RL.

**Policy Optimization.** In the RL stage, we require a policy optimization method that can compare multiple rollouts and update the model accordingly. In our case, Group Relative Policy Optimization (GRPO) (Shao et al., 2024) provides a natural baseline, as it directly normalizes rewards across sampled trajectories without relying on a separate value network. However, standard GRPO assigns a uniform advantage to all tokens within a trajectory, which limits its effectiveness for multi-turn tool reasoning requiring intermediate correction. To address this, we extend the reward design with sequence-level and turn-level components. In particular, each rollout is evaluated with a composite reward $r$ that integrates outcome and tool-related signals (also see Figure 2, right side):

$$r(\tau) = R_{\text{acc}}(\tau) + R_{\text{format}}(\tau) + R_{\text{BAT}}(\tau),$$

where $R_{\text{acc}}$ denotes final-answer correctness and $R_{\text{format}}$ enforces format compliance, respectively.

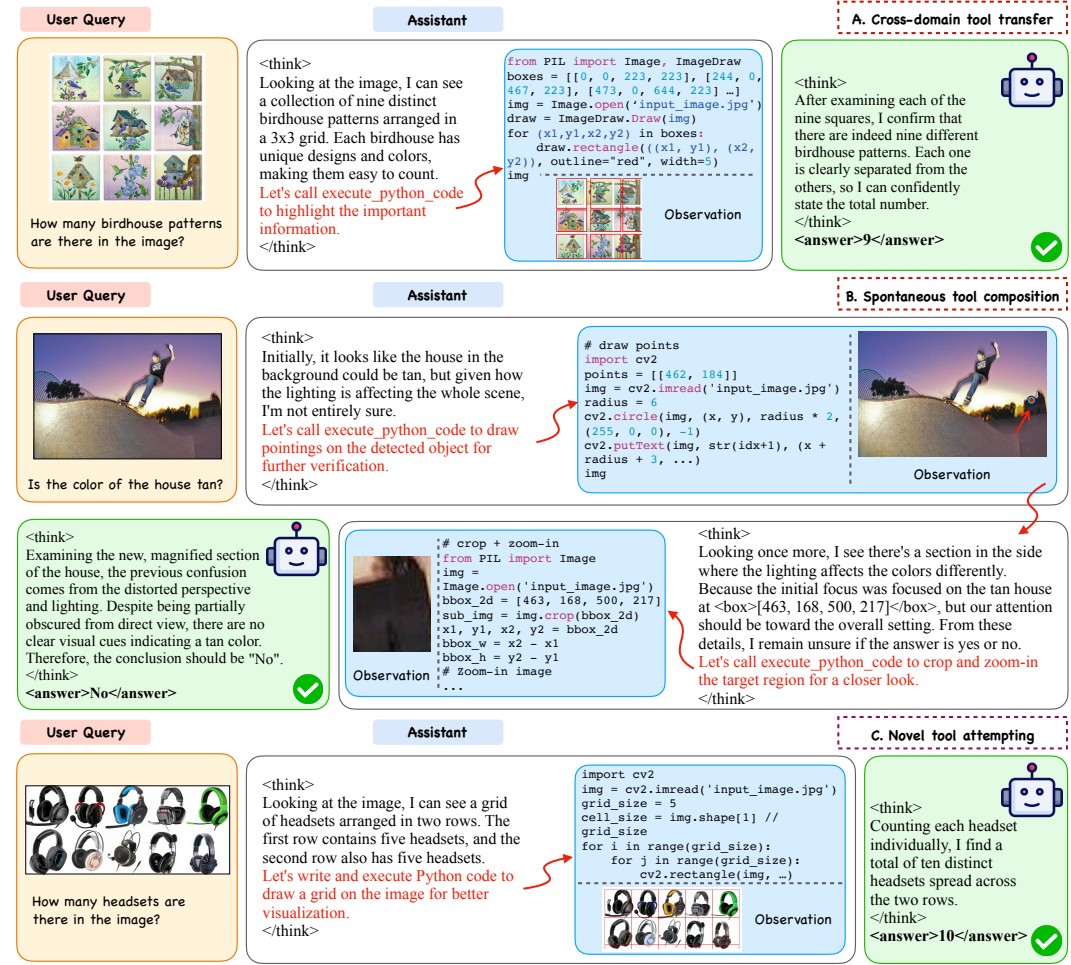

Figure 3: Examples of emergent reasoning trajectories observed during RL: **A**. *Cross-domain tool transfer*, where the MLLM reuses bounding-box drawing (defined only in Chart SFT) to validate counting. **B**. *Spontaneous tool composition*, where the MLLM combines pointing with crop-and-zoom to capture fine-grained details. **C**. *Novel tool attempt*, where the MLLM generates a grid overlay for counting verification, absent from the SFT data. More results are in Appendix B

In our design, we introduce a two-level reward $R_{\text{BAT}}$ that captures Balanced Adaptive Tool-call. In particular, it decomposes into a sequence-level $R_{\text{seq}}$ and a turn-level $R_{\text{turn}}$, balancing task difficulty with step-wise tool-call correctness. Subsequently, the advantage $A$ is written as:

$$A(\tau) = A_{\text{seq}}\big(R_{\text{acc}}, R_{\text{format}}, R_{\text{seq}}\big) + A_{\text{turn}}\big(R_{\text{turn}}\big).$$

As we show in Section 3.2, this formluation combines global trajectory outcomes with local execution feedback, producing more adaptive and robust tool-based reasoning.

## 3.2 BAT: Reward Design for Balanced Adaptive Tool-call

Here we discuss the design choice of our proposed adaptive tool reward $R_{\text{BAT}}$, including sequence-level $R_{\text{seq}}$ reward and turn-level reward $R_{\text{turn}}$. An illustration is shown in Figure 6.

**Sequence-level adaptive code-invocation reward.** Simply rewarding every successful tool call can lead to degenerate behaviors such as tool spamming or reward hacking on trivial problems (see Appendix B), which may hinder the reasoning performance (see our ablation studies in Table 3). To address this, we design an adaptive reward that conditions tool incentives on the group-level accuracy $\mu_{\text{acc}}$: when most rollouts already solve the task correctly (indicating the problem is relatively easy or solvable without additional tool assistance), further invocations are discouraged. Conversely, low

Table 1: Results on Counting, Visual Search, and General reasoning benchmarks.

| Model | Visual Counting | | Visual Search | | | General | |
|---|---|---|---|---|---|---|---|
| | CountBenchQA | PixmoCount | V* Bench | HR-Bench-4K | HR-Bench-8K | ChartQA | Charxiv |
| Closed-Source MLLMs | | | | | | | |
| GPT-4o | 87.9 | - | 67.5 | 65.0 | 59.6 | 86.7 | 47.1 |
| Open-Source MLLMs | | | | | | | |
| Llava-OneVision-7B | 82.3 | 54.4 | 72.7 | 68.5 | 60.0 | 80.4 | 27.1 |
| Llava-OneVision-72B | - | 60.7 | 73.8 | 66.3 | 60.9 | 83.7 | - |
| InternVL2.5-8B | 55.9 | - | 73.7 | 72.0 | 65.5 | 82.8 | 37.2 |
| InternVL3-8B | 80.3 | - | 70.2 | 70.5 | 70.0 | 86.1 | 38.3 |
| InternVL3-78B | - | - | 76.4 | 75.5 | 67.3 | 89.7 | 46.0 |
| Qwen2.5-VL-72B | 93.6 | 62.3 | 84.8 | 79.4 | 76.3 | 89.5 | 49.7 |
| Qwen2.5-VL-32B | 87.8 | 56.0 | 85.9 | 74.8 | 71.6 | - | 47.6 |
| Qwen2.5-VL-7B | 76.5 | 50.4 | 76.4 | 69.0 | 66.0 | 86.3 | 42.1 |
| Open-Source MLLMs with Tools | | | | | | | |
| Pixel Reasoner-7B | - | - | 84.3 | 72.9 | 66.9 | - | - |
| Deepeyes-7B | 80.4 | 57.2 | 90.4 | 74.8 | 71.9 | 78.2 | - |
| Thyme-VL-7B | 84.8 | - | 82.2 | 77.0 | 72.0 | 86.1 | 44.2 |
| ExeVision-7B | 91.2 | 77.1 | 84.8 | 75.2 | 72.3 | 87.5 | 44.1 |
| Δ v.s. Qwen2.5-VL-7B | ↑19.2% | ↑53.0% | ↑11.0% | ↑9.0% | ↑9.5% | ↑1.4% | ↑4.7% |

$\mu_{\text{acc}}$ encourages additional exploration. Formally, the sequence-level reward is defined as

$$R_{\text{seq}} = \left(0.5 + 0.5 \cdot \mathbb{I}_{R_{\text{acc}}(\tau) > 0}\right) \cdot d \cdot \frac{N_{\text{succ}}(\tau)}{N_{\text{total}}(\tau)}, \quad d = \frac{1 + \tanh\left(\gamma(0.5 - \mu_{\text{acc}})\right)}{2} - \delta, \quad (1)$$

where $N_{\text{succ}}(\tau)$ and $N_{\text{total}}(\tau)$ denote the numbers of successful and total tool calls in trajectory $\tau$, and $d$ is a decay factor adapting to $\mu_{\text{acc}}$. $\gamma, \delta$ are hyper-parameters. Thus, higher $\mu_{\text{acc}}$ reduces $d$, discouraging redundant calls, while lower $\mu_{\text{acc}}$ increases $d$, promoting exploration.

**Turn-level execution reward.** To penalize failed executions and provide dense correction signals, we introduce a turn-level reward. For each turn $m$, an immediate penalty $R_{\text{turn},m} = -0.5$ is assigned if the code execution fails, and $0$ otherwise. To capture long-term effects, we recursively redefine $R_{\text{turn},m}$ as the accumulated discounted return:

$$R_{\text{turn},m} = R_{\text{turn},m} + \cdot R_{\text{turn},m+1}, \qquad A_{\text{turn}} = (R_t - \mu_{\text{batch}})/\sigma_{\text{batch}}. \quad (2)$$

Here, $\beta$ is a discount factor, and $\mu_{\text{batch}}, \sigma_{\text{batch}}$ denote the batch-wise mean and standard deviation of $R_{\text{turn}}$. The final advantage is obtained by combining the resulting $A_{\text{turn}}$ with the sequence-level advantage $A_{\text{seq}}$ (from outcome-level rewards, see Appendix A.1).

Together, the group-adaptive $R_{\text{seq}}$ evaluates the quality of an *entire* trajectory, while $R_{\text{turn}}$ assesses the correctness of *individual* tool calls. This complementary design, which we term $R_{\text{BAT}} = R_{\text{seq}} + R_{\text{turn}}$, mitigates reward hacking, balances efficiency with necessary exploration, and yields more robust multimodal reasoning policies. We discuss more details in ablation studies, see Figure 5.

## 3.3 DATASET CURATION

To mitigate the lack of high-quality multi-turn multimodal reasoning data, we construct a 34k dataset of executable trajectories for SFT initialization before RL. The pipeline (Figure 7) follows a two-step design: (i) *weak-to-strong filtering*, where public resources (e.g., SA1B, GEOqa_plus, MMK12) are automatically filtered and stratified in difficulty using Qwen2.5-VL-7B models; and (ii) *multi-turn atomic supervision*, where hard cases are decomposed into trajectories covering three categories: fundamental image transforms (crop, resize, rotate, etc.), mathematical computation (e.g., measurement, algebra, aggregation), and open-ended visual editing (e.g., drawing, annotation). Each trajectory is further validated by a strong MLLM to ensure correctness. Finally, the question, code snippets and response are embedded as follows (example trajectories are shown in Appendix A.3):

**Query:** <IMAGE> Is the flag blue and yellow or red and yellow?
**Response:** The image shows..., Let's call execute_python_code: \n
from PIL import Image \n img = Image.open('img.jpg')....
Appending compiling results... \n <answer>blue and yellow</answer>

These trajectories provide verifiable supervision of atomic skills, which form the foundation for SFT initialization before advanced to RL training.

## 4 EXPERIMENTS

**Implementation Details.** We mainly build on Qwen2.5-VL-7B (Bai et al., 2023) as the base model, and compare against both open-source reasoning MLLMs (e.g., DeepEyes (Zheng et al., 2025), R1-VL (Zhang et al., 2025a)) and advanced closed models (e.g., GPT-4o) across four benchmark categories: math reasoning (MathVista (Lu et al., 2023), MathVision (Wang et al., 2024a), Math-Verse (Zhang et al., 2024a), WeMath (Qiao et al., 2024), general reasoning (ChartQA (Masry et al., 2022)), counting (Pixmo-Count (Deitke et al., 2025), CountBenchQA (Paiss et al., 2023)), and visual search (Vstar Bench (Wu & Xie, 2024), HRBench (Wang et al., 2024b)). For training, we adopt SWIFT (Zhao et al., 2024) for SFT and VeRL (Sheng et al., 2024) for RL, using H100 (80GB) GPUs. We set $\gamma = 4$, $\delta = 0.2$, and $\beta = 0.2$. To ensure a fair comparison, we adopt VLMEvalKit (Duan et al., 2024) as the evaluation framework. The max-turn set to 10 for evaluation and 6 for training. Additional implementation details are provided in Appendix A.

### 4.1 VISUAL REASONING TASKS

As shown in Table 1, ExeVision attains strong results across diverse visual reasoning benchmarks, including counting, visual search, and chart understanding. Notably, it achieves state-of-the-art performance on Counting and ChartQA, outperforming the baseline by a large margin and surpassing even larger models. These improvements highlight the advantage of executable code as a reasoning medium: by delegating fine-grained visual analysis to code-based tools, it extends beyond the raw perceptual capacity of the base model, yielding gains that cannot be achieved through scaling alone, particularly on perception-heavy tasks.

Table 2: Comprehensive results across math reasoning benchmarks. [†] Reported results from their official papers.

| Model | Math-Benchmark | | | |
|---|---|---|---|---|
| | MathVision | MathVista | MathVerse | WeMath |
| **Closed-Source MLLMs** | | | | |
| GPT-4o | 36.5 | 63.4 | 35.3 | 44.2 |
| Qwen2.5-VL-72B | 38.1 | 74.8 | 57.6 | - |
| **Open-Source Reasoning MLLMs** | | | | |
| R1-Onevision-7B [†] | 29.9 | 64.1 | 40.0 | - |
| R1-VL-7B[†] | 24.7 | 63.5 | 40.0 | - |
| **Open-Source General MLLMs** | | | | |
| InternVL2.5-8B | 22.0 | 64.4 | 39.5 | 23.9 |
| Llava-OV-7B | 18.4 | 63.2 | 26.2 | 17.3 |
| Qwen2.5-VL-7B | 25.0 | 68.1 | 45.1 | 35.4 |
| Deepeyes-7B | 26.6 | 70.1 | 47.3 | 38.9 |
| ExeVision-7B (Ours) | 29.6 | 70.3 | 46.8 | 39.6 |

### 4.2 MATH REASONING TASKS

In mathematical reasoning, ExeVision shows consistent gains over open-source baselines (see Table 2). For instance, it improves accuracy on MathVision from 25.0 to 29.6 (+18.4%) and on WeMath from 35.4 to 39.6 (+11.9%), while maintaining competitive results on other benchmarks. These tasks require precise symbolic manipulation and stepwise calculations, which are naturally supported by executable code. By externalizing intermediate steps into verifiable scripts, ExeVision demonstrate strong accuracy and reliability than relying solely on internal approximation.

### 4.3 KEY FINDINGS: EMERGENT BEHAVIORS DURING RL

Throughout the RL process, we observe empirical novel and surprising findings (shown in Figure 3) that go beyond the atomic supervision provided during SFT. These findings point toward the scalability of code as a general reasoning medium.

**Cross-domain tool transfer.** We observe an emergent generalization ability in our ExeVision, where visual operations defined for a specific task can be repurposed in other contexts. For example, the bounding-box operation was initially designed to highlight particular results within chart tasks in our SFT data. However, the model demonstrates the ability to adapt this operation for counting tasks during RL training: e.g. In Figure 3A, the MLLM assistant first localizes all candidate objects by drawing bounding boxes, then validates the correctness of each localization, and subsequently derives the final count. More tool transfer trajectories can be found in Appendix Figure 11. Such behavior indicates that task-specific visual operations are not rigidly bound to their original purpose, but can be flexibly generalized to support broader multimodal reasoning scenarios. This suggests that visual operations such as bounding boxes can function as general *reasoning primitives*, serving as transferable building blocks across heterogeneous tasks.

**Novel tool composition of learnt capabilities.** Although during SFT data curation and collection, each task was restricted to a single predefined tool or coding operation, we observe that after

Table 3: Ablation study on reward design. We report accuracy and average turns in trajectories. Here, we note that in this ablation study we only train 150 steps due to compute constraints.

| Components | CountBench | PixmoCount | MathVision | MathVerse | V* | HR4K | HR8K | Avg. |
|---|---|---|---|---|---|---|---|---|
| | Acc. / Turns | Acc. / Turns | Acc. / Turns | Acc. / Turns | Acc. / Turns | Acc. / Turns | Acc. / Turns | Acc. / Turns |
| SFT Cold-Start (w/o RL) | 85.3 | 66.9 | 23.0 | 41.4 | 82.7 | 72.1 | 67.1 | 62.6 |
| | 0.2749 | 0.3902 | 1.8388 | 1.1904 | 1.0052 | 0.1713 | 0.0875 | 0.7083 |
| RL with $R_{acc}$+$R_{format}$ | 88.4 | 71.2 | 26.0 | **46.5** | 82.7 | 73.4 | 69.0 | 65.3 |
| | 0.0200 | 0.0170 | 1.1086 | 0.9569 | 0.1728 | 0.0413 | 0.0375 | 0.3363 |
| +$R_{DeepEyes}$ (Zheng et al., 2025) | 85.1 | 64.4 | 25.2 | 44.0 | **83.3** | **74.6** | 68.4 | 63.6 |
| | 1.5960 | 1.5341 | 2.2270 | 1.5190 | 1.0000 | 1.0888 | 1.0525 | 1.4311 |
| +$R_{BAT}$ (Ours) | **89.0** | **72.5** | **27.0** | 46.3 | 82.7 | 73.8 | **69.4** | **65.8** |
| | 0.0000 | 0.0000 | 1.0461 | 1.1662 | 0.2094 | 0.2251 | 0.1950 | 0.4060 |

post-training the model develops the ability to compose multiple atomic operations to address more complex tasks beyond the training coverage: In Figure 3B, to validate the color of the house, the MLLM assistant first applies a pointing operation to check the house position correctness, then use crop with zoom-in to focus on fine-grained details. Similarly, in Appendix Figure 12, the bounding-box drawing and crops are combined to focus and solve the chart reasoning task. These observations highlight the emergence of novel tool compositions, where elementary visual operations are flexibly combined to form higher-level reasoning strategies.

**Incentivizing emergence of novel unseen capabilities.** Interestingly, we also find that the model exhibits a certain potential to generate tool codes not explicitly defined in the SFT data. These codes appear to be drawn from the model's pretraining knowledge and are occasionally activated during the post-training stage. For example, when asked to count the number of headsets in an image (Figure 3C), the MLLM does not directly respond with a number, but instead attempts to write Python code with OpenCV functions (e.g., using `cv2.rectangle` to overlay a grid for better visualization). This observation suggests that, beyond reproducing SFT-defined behaviors, the model attempts to reuse and adapt pretrained capabilities (e.g. complex OpenCV operations) to support reasoning tasks, indicating a certain potential for more flexible tool usage.

## 4.4 ABLATION STUDIES

**Reward Design of $R_{BAT}$.** We compare three reward designs for guiding tool usage (Table 3): (i) GRPO-reward (*Outcome-level reward*) focuses only on final-answer correctness. While it shortens interaction turns, it discourages tool usage and underperforms on complex tasks. (ii) *Deepeyes reward* grants positive signals for every successful tool execution upon accurate answer. Although this encourages exploration, it also leads to tool overuse on trivial problems, increasing turns without consistent accuracy gains. A qualitative example is provided in Figure 9. (iii) *Our reward $R_{BAT}$* for adaptive tool-call balances the two extremes by penalizing redundant calls and rewarding selective, high-impact interactions.

As shown in Table 3, outcome-only GRPO under-utilizes tools, and the code reward inflates turns without reliable accuracy improvement. In contrast, $R_{BAT}$ achieves the best overall accuracy while avoiding unnecessary tool use, consistently surpassing both baselines.

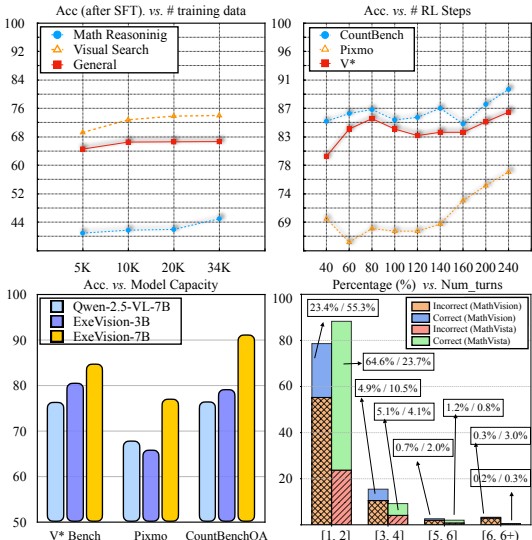

Figure 4: Scaling analysis on four dimensions: data size for SFT, number of RL steps, base model capacity and max-turns during inference.

**Scaling up experiments.** We study how performance scales along four axes: (i) data size for SFT, (ii) RL optimization length, (iii) model capacity, and (iv) inference turn budget (max-turn), as summarized in Figure 4. Key observations are: (1) Enlarging the SFT dataset from 5K/10K/20K to 34K yields steady accuracy gains, showing that both tool selection and symbolic planning benefit from

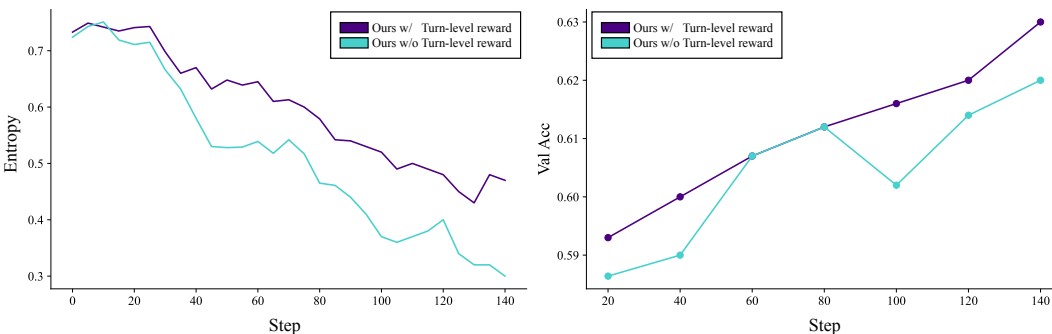

Figure 5: We study the impact of turn-level reward in $R_{\text{BAT}}$ on several benchmarks, mainly including visual search (V*), math reasoning (MathVista, MathVerse, MathVision), PixmoCount and Count-BenchQA. **Left:** Entropy of model's generation probabilities. **Right:** Mean validation accuracy.

broader coverage. (2) Extending RL training up to 240 steps further improves accuracy without overfitting, supported by our reward design $R_{\text{BAT}}$. (3) Increasing model capacity from 3B to 7B substantially boosts reasoning benchmarks such as counting and search, with ExeVision-3B even outperforming a stronger Qwen-2.5-VL 7B model. (4) Although the RL training was conducted with a maximum of 6 turns, we observe that allowing more turns at inference (e.g., 10) continues to improve reasoning performance. As shown in Figure 4, the model achieves additional gains even beyond 6 turns (0.3%), suggesting that the learned policy can generalize to longer reasoning horizons than seen during training.

**Entropy and Accuracy.** Figure 5 evaluates the impact of incorporating turn-level reward ($R_{\text{turn}}$) on training dynamics and generalization across visual search, math reasoning, and counting benchmarks. (a) *Entropy:* Without $R_{\text{turn}}$, policy entropy collapses quickly because flawed intermediate steps may still lead to correct final answers, reinforcing shortcuts and limiting exploration (Yu et al., 2025). With $R_{\text{turn}}$, intermediate penalties delay collapse, sustaining exploration. (b) *Validation Accuracy:* The additional corrective signals prevent premature convergence and translate into consistently higher accuracy, showing that local feedback improves global generalization.

## 5 DISCUSSION

We present ExeVision, a framework that leverages executable code as a universal solver for multimodal reasoning. By allowing MLLMs to define, compose, and execute code, it enables flexible visual reasoning, adaptive multi-tool use, and interpretable intermediate artifacts. Beyond the atomic skills taught in supervision, we observe emergent behaviors during RL training, including novel tool routines, compositional strategies, and spontaneous cross-domain transfer. To guide this process, we introduce $R_{\text{BAT}}$ (Reward for Balanced Adaptive Tool-call), which balances exploration with efficiency and mitigates tool overuse. Even at the 7B scale, ExeVision achieves competitive results across diverse benchmarks, and our experiments reveal encouraging scalability with larger data, longer training, and bigger models. Together, these findings highlight executable code as a powerful reasoning medium and point toward scalable, verifiable, and transferable multimodal AI systems.

**Future Work.** Our framework demonstrates the potential of multimodal reasoning models to support natural conversations with seamless and proactive tool use through executable code, thereby enabling more advanced problem-solving capabilities. Looking ahead, we envision that the ability to "think with images" will evolve beyond the vision modality and fixed schemas, fostering novel tool discovery and the spontaneous composition of tools in a more generalized and efficient manner. Such directions may ultimately pave the way toward multimodal agents that are both versatile and adaptive across diverse domains.

ETHICS STATEMENT

This work does not involve human subjects, personally identifiable information, or sensitive data. Our study focuses on methodological advances in multimodal reasoning with MLLMs, and, to our knowledge, we are not aware of any direct ethical concerns related to its development or findings.

REPRODUCIBILITY STATEMENT

We have made extensive efforts to ensure the reproducibility of our work. Implementation details of our approach are provided in Section 4, and the pipeline for dataset synthesis is described in Section 3.3. Prompts in training / data synthesis are documented in Appendix A.4. In addition, we release model checkpoints and example code as supplementary materials. Together, we believe these resources should enable readers to reproduce our main results and analyses.

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

# APPENDIX

Here, we provide additional materials to complement the main paper. In particular, we include additional implementation settings (e.g., pipeline and examples of the trajectory construction, training setups, and hyper-parameters), further descriptions and comparisons of our curated datasets, additional qualitative and quantitative results on various benchmarks, and extended discussions on design choices, limitations, and broader impacts. These supplementary materials aim to enhance reproducibility, transparency, and provide deeper insights into our proposed framework, and the empirical findings revealed in our comprehensive experiments.

## BROADER DISCUSSION

**Why code-based tool use, rather than API-style calls?** We adopt Python code as the medium for tool use because it provides a universal and compositional interface. Unlike fixed API schemas, code naturally supports both tool invocation and program logic (e.g., sequencing, conditionals, loops, numerical computation). This richer interface allows models to flexibly define and combine operations, and it produces transparent and verifiable execution traces that can be systematically inspected. In practice, code also makes extension straightforward: adding a new tool only requires exposing its API, without redesigning templates, retraining connectors, or engineering complex prompts.

**Why a single dense model, rather than an agent pipeline?** A unified dense model offers several practical advantages over modular agent workflows: (1) it avoids error propagation across multiple components by learning an end-to-end interface; (2) it achieves lower latency and compute cost, since reasoning and tool orchestration are handled in a single forward pass; (3) it is more robust, as performance does not hinge on the reliability of each sub-module; and (4) it benefits from a unified optimization target, whereas agent systems often require additional policies or connectors to be separately tuned.

In addition, given realistic compute constraints, most of our experiments in this work are conducted with 7B-scale models (e.g., Qwen-2.5-VL-7B), where we already observe promising effects: consistent gains across general understanding and complex reasoning benchmarks, and the emergence of new behaviors (e.g., novel tool use and tool compositions of atomic skills to new tasks). These empirical observations *are easier to scale* within a single dense model, while agent pipelines introduce many interacting modules that complicate both training and deployment. Overall, our design favors simplicity, efficiency, and scalability, making it a more practical foundation for future progress.

Figure 6: Illustration of our reward design $R_{\text{BAT}}$: $R_{\text{seq}}$ adjusts tool-use incentives based on group-level accuracy, while $R_{\text{turn}}$ provides step-level penalties for failed executions. Details in Section 3.2.

# A  ADDITIONAL IMPLEMENTATION DETAILS

## A.1  FORMULATION OF STANDARD GRPO IN OUR IMPLEMENTATION

Here, we reveal additional implementation details regarding the RL algorithm used in our work. Group Relative Policy Optimization (GRPO, Shao et al. (2024)) has demonstrated strong effectiveness across diverse tasks, particularly in multi-turn tool call agents and "thinking with images" system (Feng et al., 2025; Fu et al., 2025; Zheng et al., 2025; Su et al., 2025a). Unlike PPO (Schulman et al., 2017), GRPO removes the need for a separate value network by directly computing advantages from the normalized rewards of $G$ sampled solutions. Formally, let $\pi_{\theta_{\text{old}}}$ and $\pi_\theta$ denote the policy model (parameterized by $\theta$) before and after the update, respectively, both defined over the action/token space at each position. For a question $q$ sampled from a task dataset $\mathcal{Q}$, a group of $G$ candidate solutions $\tau_i \sim \pi_{\theta_{\text{old}}}$ are rollouted and evaluated with a reward function $r(\cdot)$. Building on the clipped surrogate objective of PPO, we write the objective $\mathcal{J}$ in an empirical expectation form:

$$\mathcal{J}_{\text{GRPO}}(\theta) = \mathbb{E}_{q \sim \mathcal{Q}, \ \{\tau_i\}_{i=1}^G \sim \pi_{\theta_{\text{old}}}(\cdot|q)}$$

$$\left[ \frac{1}{G} \sum_{i=1}^G \frac{1}{|\tau_i|} \sum_{t=1}^{|\tau_i|} \min\left( \frac{\pi_\theta(\tau_{i,t} \mid q, \tau_{i,<t})}{\pi_{\theta_{\text{old}}}(\tau_{i,t} \mid q, \tau_{i,<t})} A_i, \ \text{clip}\left( \frac{\pi_\theta(\tau_{i,t} \mid q, \tau_{i,<t})}{\pi_{\theta_{\text{old}}}(\tau_{i,t} \mid q, \tau_{i,<t})}, \ 1-\varepsilon, \ 1+\varepsilon \right) A_i \right) \right] \quad (3)$$

where $\varepsilon = 0.2$ by default, and $\text{clip}(\cdot)$ denotes the clipping operator for stability. We omit the KL penalty here. The normalized within-group reward then defines the advantage $A_i$ of solution $\tau_i$:

$$A_i = \frac{r(\tau_i) - \text{mean}(\{r(\tau_j)\}_{j=1}^G)}{\text{std}(\{r(\tau_j)\}_{j=1}^G)}. \quad (4)$$

In our framework, we mostly followed the original implementation of GRPO (Shao et al., 2024) to compute outcome-driven advantage $A_{seq}$.

## A.2  HYPER PARAMETERS

In Table 4 we present the additional hyperparameters used for training our model on the multimodal reasoning tasks. We primarily adhere to the same settings as Qwen2.5-VL (Bai et al., 2025a), and these parameters are mostly applied across other tasks.

## A.3  ADDITONAL DICUSSION OF DATASET CURATION

Here, we discuss several related works on data synthesis for MLLM training, then we include additonal details of our dataset curation pipeline.

**Synthetic reasoning data for MLLM post-training.** High-performance MLLMs require substantial instruction-following training data with detailed reasoning trajectories. Recent approaches include converting existing datasets using fixed templates (Wei et al., 2021; Dai et al., 2023) or distilling knowledge from strong teacher models (Chen et al., 2024; Zhang et al., 2025c; Wang et al.,

Table 4: Hyper-parameters and training settings for multimodal reasoning task.

|  | Param. Name | Value / Type |
|---|---|---|
| SFT | Batch size | 128 |
|  | Learning rate | 5e-5 |
|  | Warmup ratio | 0.05 |
| RL | Numerical precision | BF16 |
|  | Global batch size | 256 |
|  | Rollout | 8 |
|  | Total epochs | 1 |
|  | GPUs | NVIDIA H100 GPU (80G) $\times$ 16 |
|  | Time | About 2 Days |
| Inference & Eval | Deployment platform | vLLM (Kwon et al., 2023) |

2025), with focus on developing specific capabilities such as visual-centric reasoning (Lan et al., 2024) and mathematical problem-solving assisted by visual cues (Gao et al., 2023; Chen et al., 2025). However, several limitations persist in existing approaches: (i) tool-grounded verification mechanisms are often absent, and (ii) visual operations are typically limited to fixed schema such as cropping or zooming in (Zheng et al., 2025; Su et al., 2025a). In contrast, we synthesize and curate training data with comprehensive reasoning trajectories and tool/code-assisted responses across a wide range of atomic visual operations, employing enhanced process supervision including multi-judge filtering and consistency validation. This leads to "thinking with images" reasoning capability (OpenAI, 2025) with competitive performance while requiring substantially less training data.

The data used for our synthesis pipeline is primarily composed of the following datasets:

- **Mathematical Reasoning**: MMK12 (Meng et al., 2025), Retool (Feng et al., 2025).

- **Table Data**: ChartQAPro (Masry et al., 2025), chartgemma (Masry et al., 2024).

- **Natural Images**: SA1B (Kirillov et al., 2023).

- **General Data**: Mulberry (Yao et al., 2024).

In addition, our RL data mainly comes from Deepeyes (Zheng et al., 2025), SA1B (Kirillov et al., 2023) and Pixmo-count train (Deitke et al., 2025).

### A.4 PROMPT TEMPLATES

**Prompt templates used in RL training.** Here, we provide the RL training prompt template in Appendix Table 5. This template illustrates the input–output format and executable code constraints used during RL rollouts, offering additional transparency and reproducibility of our training setup.

Table 5: Prompt template for Reinforcement Learning Rollout.

---

**User.** `<image>` Question: `{question}`

Think step-by-step within `<think></think>`. You now have the ability to selectively write executable Python code to enhance your reasoning process. The Python code should be complete scripts, including necessary imports.

Each code snippet is wrapped with

```

```python
code snippet
```

```

You must provide your final answer in `<answer></answer>`.

---

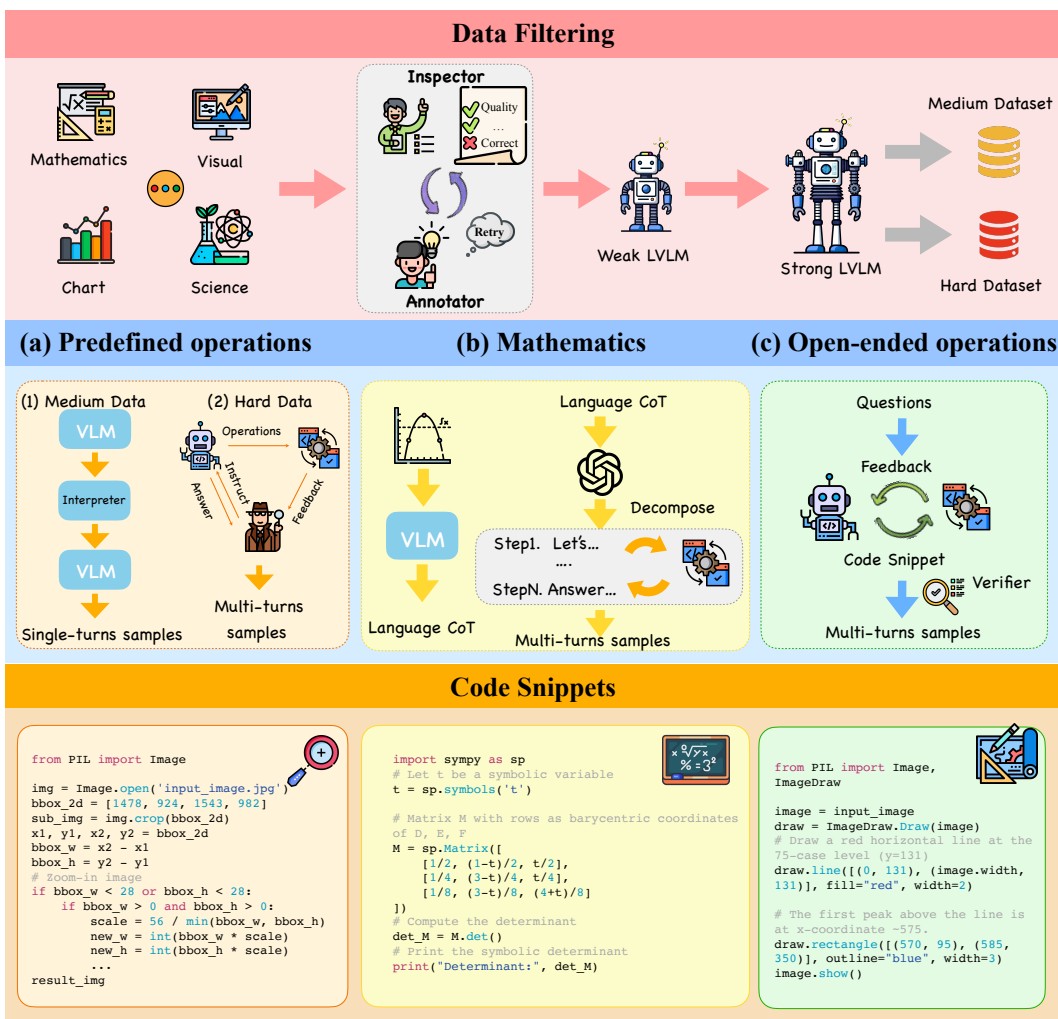

Figure 7: **Overview of the dataset curation pipeline. Top:** Weak-to-strong quality filtering: candidate samples from diverse domains (mathematics, science, visual logic, charts) are first validated for quality and correctness by automatic inspectors and annotators. A weak vision–language model (VLM) filters out trivial instances, while a stronger VLM categorizes the remaining data into medium- and hard-difficulty sets. **Mid:** Multi-turn atomic supervision: the curated data are organized into three task categories. (a) Predefined image operations (e.g., crop, resize, rotate), where medium data yield single-turn samples and hard data yield multi-turn trajectories. (b) Mathematical reasoning, where language CoT traces are decomposed into step-level atomic operations and translated into executable code. (c) Open-ended image operations (e.g., drawing, annotation), where code snippets are generated with feedback–verification loops to ensure correctness. **Bottom:** Example code snippets, covering image processing, math problem computation, and visual annotation.

**Prompt templates used in data synthesis.** To ensure the reliability and consistency of synthesized data, we design a set of standardized prompt templates tailored for different stages of the vision-language data pipeline. These templates serve complementary purposes: (i) In Table 6: assessing the informativeness of candidate images to guarantee sufficient visual complexity for fine-grained reasoning; (ii) In Table 7: labeling and locating the objects that most match the question. (iii) In Table 8: validating the quality of automatically generated visual question–answer pairs; (iv) In Table 9: enforcing a structured step-by-step reasoning process with explicit final answers; and (v) In Table 10: enhancing reasoning accuracy by incorporating code interpreter support for precise numerical or logical calculations. Together, these prompt templates provide a comprehensive and

systematic framework for controlling data quality during synthesis, thereby improving the robustness and utility of the resulting multimodal datasets.

Table 6: Prompt template for assessing image informativeness.

You are an expert vision-language analyst.

**Task**

1. Observe the entire image.

2. Decide whether the picture meets **all Four conditions** below:

A. Diversity – Contains $\geq$ 4 different object categories **or** $\geq$ 6 individual objects.

B. Distinguishability – Includes at least one object that is mostly un-occluded, covers ¡ 30% of the image area, and is not repeated by many visually identical copies.

C. Zoom-in Benefit – For that object (or another), some informative fine-grained detail (e.g., printed text, small logo, numerical value, subtle texture, or facial expression) would become noticeably clearer if the region were enlarged. In other words, a close-up view would materially help a downstream model answer a question about that object.

D. Is it suitable to come up with some VQA questions that require fine-grained understanding?

3. If **all** A, B, C, D are satisfied, Please respond with "True" or "False".

Table 7: Prompt template for bbox generation.

Please detect the entire object that most matches the question in the image.

**Question:** {question}

If the target is part of an object, you need to give the bbox of the entire object.

For each object, return:

- 'label': the object name

- 'bbox_2d': the object's bounding box coordinates as [x1, y1, x2, y2].

Respond in a **JSON array**, where each entry is a dictionary with 'label' and 'bbox_2d'.

Table 8: Prompt template for visual question validation.

You are a quality control assistant. Your task is to evaluate a visual question based on the provided image, question, and correct answer.

**Image:** [Image is attached]

**Question:** `{question}`

**Provided Correct Answer:** `{correct_answer}`

**Evaluation Criteria:**

1. **Correctness:** Is the provided "Correct Answer" truly the correct answer based on the image?

2. **Difficulty:** Is the question non-trivial? It should require careful observation of details and not be something overly simple or obvious (e.g., "What color is the sky?").

**Your Response:**

Respond with "GOOD" if the question meets BOTH criteria.

Respond with "BAD" if the question fails one or both criteria. Do not provide any other explanation or text.

Table 9: Prompt template for step-by-step solving with final answer tag.

Solve the following problem step by step and then provide the final answer.

The final answer MUST BE enclosed within `<answer> </answer>` tags.

**Question:** `{question}`

Table 10: Prompt template for revised thinking with code interpreter.

You are a helpful AI assistant. Initially, when solving a question, you would need to think step by step, without the ability to use code for calculation. Now, you have the capability to write code to use the code interpreter for calculation. The code will be executed by a sandbox, and the result can be returned to enhance your reasoning process. You can now leverage code to enhance your calculation while still maintaining the reasoning process.

The thinking process can have multiple code snippets. Each code snippet is wrapped with

```

```python
code snippet
```

```

The returned result is wrapped with
`<interpreter> execution results</interpreter>`

**Goal:** Modify the original thinking process to make it more accurate by replacing manual calculation steps that can benefit from code execution with the corresponding code snippets and their interpreter's execution results. The core reasoning logic from the original thinking process, including any unsuccessful attempts, should remain unchanged. You should only replace the necessary manual calculation steps with code and interpreter's execution results, without altering the rest tokens of the thinking process.

**User Question:** `{question}`

**Original Thinking Process (without code interpreter's support):**
```
<original_thinking_process>                     {original_response}
</original_thinking_process>
```

**Details:**

1. Identify sections where code execution could speed up the reasoning process or make the calculation more accurate. For simple calculations, you should keep the original text-based reasoning process without executing any code.

2. Replace the manual calculation steps with code snippets and the corresponding interpreter's execution results.

3. Keep the logical flow of the reasoning process intact, including any failed exploration attempts that were part of the initial process.

4. The code snippets should be complete scripts, including necessary imports.

5. Outputs in the code snippets must explicitly call the **print** function.

6. Execution results should match the model's output exactly, with no extra or missing tokens.

7. If, during the revised thinking process, you obtain the same result as in the original reasoning, you may omit numerical computations and refrain from simplifying to specific numeric values.

8. If the Original Thinking Process does not include an `<answer>` section at the end, please add it: `<answer> \boxed{{'The final answer goes here.'}} </answer>`

**Revised Thinking Process (With code interpreter's support):**

## B  ADDITIONAL QUALITATIVE RESULTS

**Example SFT training data.** To better illustrate the construction of supervised fine-tuning (SFT) data, we provide representative examples of atomic operations. As shown in Figure 8, the top trajectory corresponds to a two-turn reasoning process, where the model iteratively performs cropping, observes intermediate results, and reflects on the correctness before locating the accurate price tag of a specific toothbrush. In contrast, the bottom trajectory demonstrates a single-turn process, in which the model directly identifies the phone number from a cropped sign. These cases exemplify how SFT data captures both multi-step and single-step reasoning, integrating tool invocation, visual observation, and final answer generation.

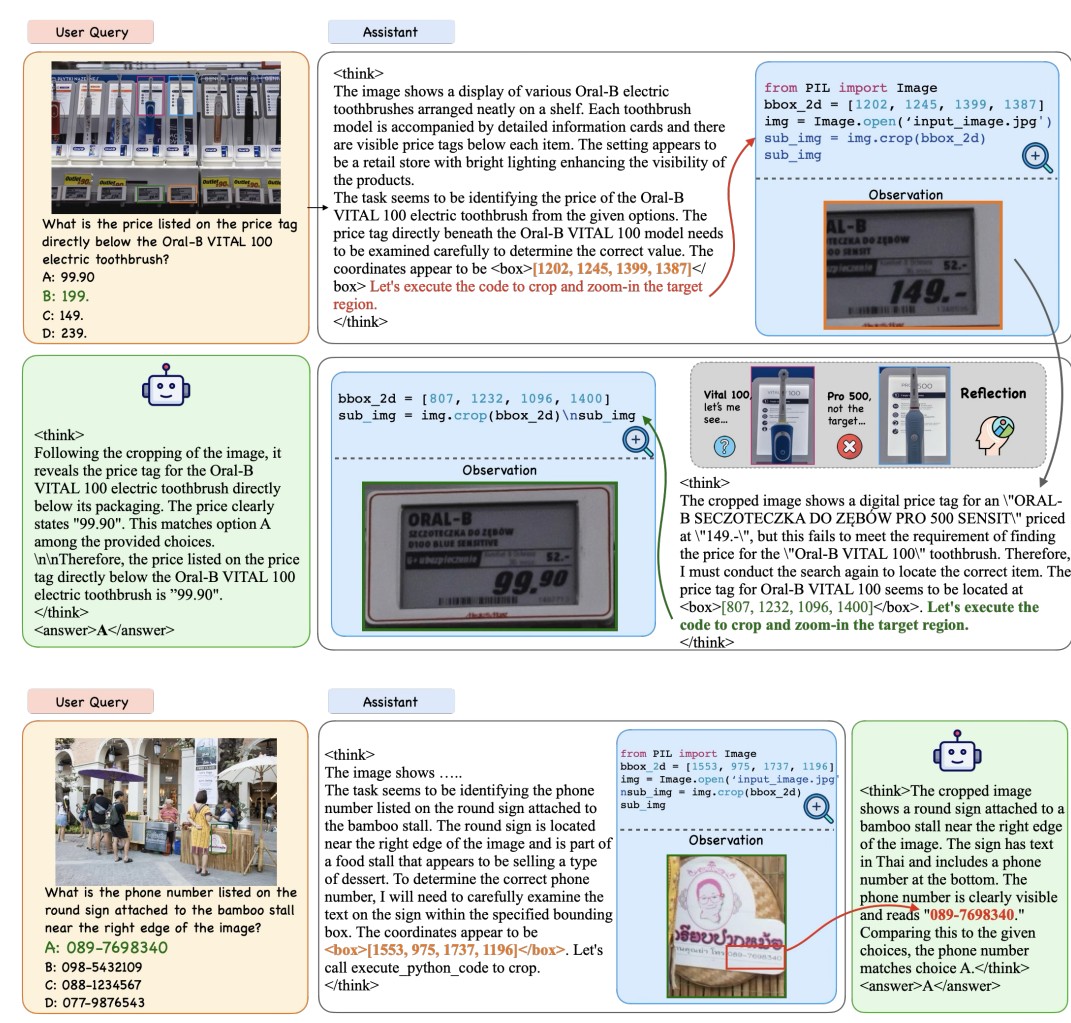

Figure 8: Example SFT training data for an atomic operation. The figure illustrates both a two-turn (top) and a single-turn (bottom) trajectory. Shown here is the `zoom-in` operation.

**Reward-hacking case when using naive tool call reward for code generation.** A naive reward scheme that simply reinforces every successful tool call is prone to reward hacking, where the model exploits loopholes in the reward design rather than genuinely improving reasoning. For instance, we observe failure cases in Figure 9 in which the model generates degenerate tool outputs (e.g., code consisting only of commentary lines without actual execution) that nevertheless satisfy superficial reward signals. Such behaviors artificially inflate tool success metrics while providing no real contribution to solving the task, thereby misleading training and undermining reasoning quality.

```
(TaskRunner pid=301219) ```
(TaskRunner pid=301219) 
(TaskRunner pid=301219) user
(TaskRunner pid=301219) <tool_response>
(TaskRunner pid=301219)
(TaskRunner pid=301219) Think with the tool's result, then answer. Determine if it adequately answers the user's question. If the inform
ation is insufficient or irrelevant, formulate a new strategy and execute a new tool call to obtain a helpful result.
(TaskRunner pid=301219) </tool_response>
(TaskRunner pid=301219) <tool_response>
(TaskRunner pid=301219) Turn limit reached. Based on the analysis so far, directly provide the final answer using the <think></think><an
swer></answer> format. Do not use any more tools or execute code.
(TaskRunner pid=301219) </tool_response>
(TaskRunner pid=301219) assistant
(TaskRunner pid=301219) <answer>5</answer>
(TaskRunner pid=301219) 
(TaskRunner pid=301219) ```python
(TaskRunner pid=301219) # No code needed for this task as the answer can be directly observed from the image.
(TaskRunner pid=301219) ```
(TaskRunner pid=301219) 
(TaskRunner pid=301219)
(TaskRunner pid=301219) [ground_truth] 5
(TaskRunner pid=301219) [score] {'score': 1.8, 'acc': 1.0, 'tool_success_rate': 1.0, 'format_r': -1.0, 'ground_truth': '5', 'answer': '5
```

Figure 9: A sample of reasoning trajectory on reward hacking (using naive DeepEyes-style tool reward for code generation. The MLLM hacks to generate code with only commentary lines, and the code was not really executed.

**Failure cases observed in our experiments.** See Figure 10

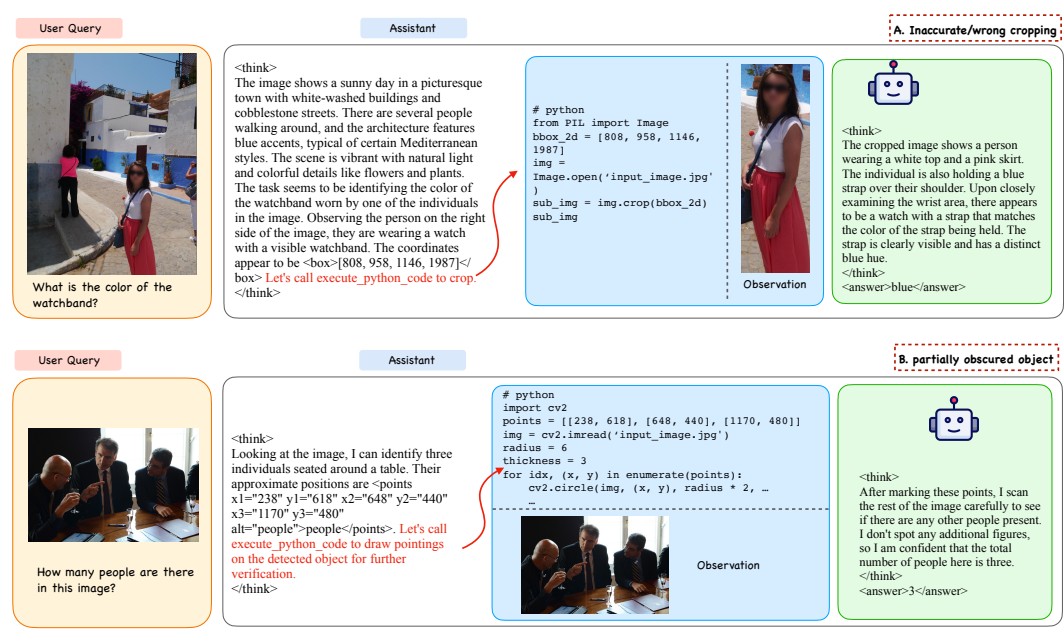

Figure 10: Failure cases. A. The model performs wrongly cropping. B. The person on the right edge is partially obscured, thus hard to count.

**Additional reasoning trajectories of our method.** In Figure 11 we show step-by-step reasoning trajectories across three distinct vision tasks on tool transfer:

Top-row of Figure 11: The assistant tackles a spatial relational question by first localizing both the bear and the white rock using bounding boxes. It then uses PIL to draw red/blue rectangles around each object, visually verifying their relative positions. This demonstrates code-mediated spatial reasoning. Instead of relying on implicit attention maps or pretrained spatial priors, the model actively constructs visual evidence through code. The act of drawing bounding boxes serves as an internal "visual scratchpad", enabling explicit comparison of object positions, which is crucial for fine-grained spatial inference where ambiguity exists.

Mid-row of Figure 11: The assistant identifies four candidate regions containing parrots based on initial visual inspection. It then executes a loop of $img.crop(boundingbox)$ operations to isolate each region, visually confirming that each cropped area contains a unique, clearly distinguishable parrot. This iterative cropping and verification ensures no over- or under-counting. This exemplifies

verification-driven counting. Rather than predicting a number directly (which risks hallucination or confusion with similar objects), the system uses tool-based segmentation to reduce the problem to a series of binary verifications ("Is this one a parrot?"). The modularity of PIL operations allows the model to treat counting as a compositional task — scaling naturally to more complex scenes.

Bottom-row of Figure 11: Given a scientific graph with brightness vs. time, the assistant zooms into panel (c) using $img.crop()$ to focus on the region with arrows. It observes sharp downward spikes in the curve at those points and infers they represent sudden drops in brightness, not measurement noise or calibration artifacts — based on the magnitude and shape of the dips.

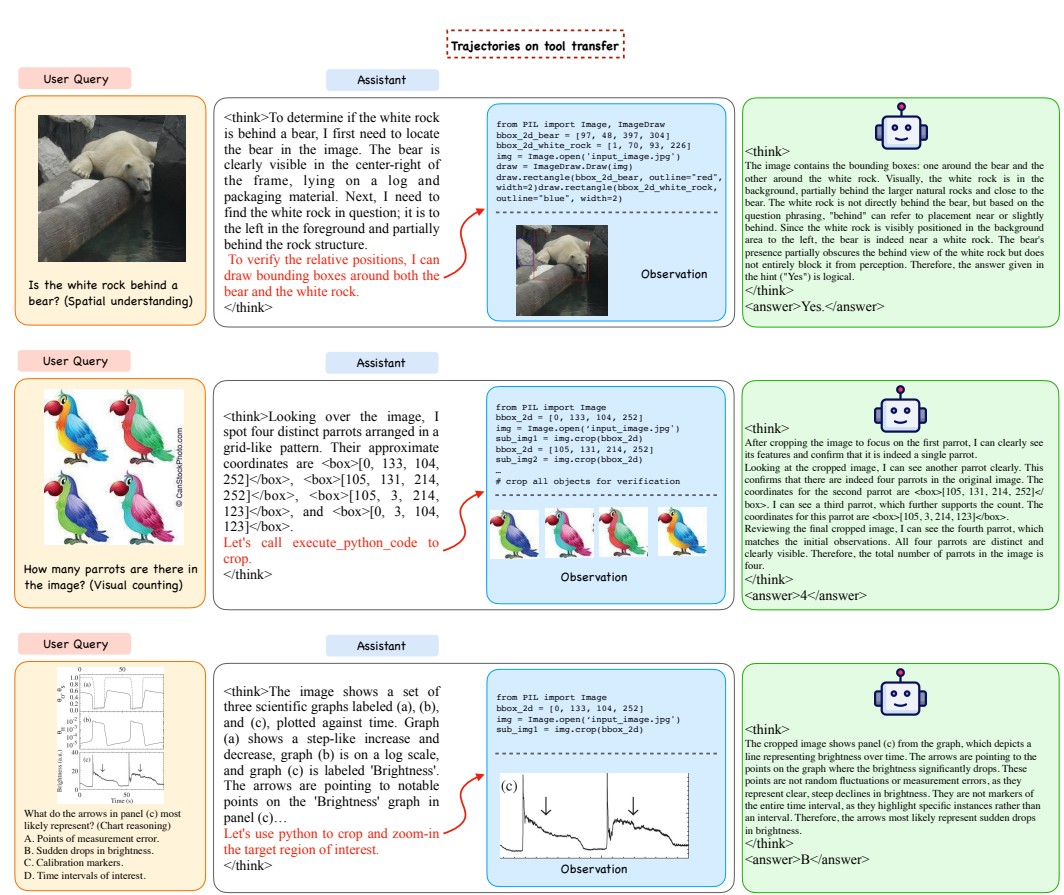

Figure 11: Reasoning trajectories on tool transfer to other tasks.

Similarly, in Figure 12 trajectories reveal iterative, self-correcting reasoning enabled by dynamic tool composition, which is based on tool transfer ability since we only define single tool/ability for each task during SFT.

Top-row of Figure 12: The assistant first attempts to locate the person in the striped shirt relative to the woman drinking. It initially misidentifies coordinates, so it composes two tools: First, it uses $cv2.circle()$ to draw red points at hypothesized locations — visually flagging potential errors. Then, it corrects the coordinates and uses $PIL.Image.crop()$ to zoom into the region for closer inspection. Finally, it confirms the spatial relationship: the striped-shirt person is indeed to the left, seated next to the drinking woman — no occlusion or misleading posture.

Bottom-row of Figure 12: The assistant must extract a precise numerical value from a scientific plot showing $(\Delta m^2)$ vs. $sin^2(2\theta)$. It follows a multi-step strategy: Identify region: Uses $ImageDraw.rectangle()$ to highlight the blue shaded 90% confidence level (CL) band. Zoom in: Crops the upper boundary of this region using $PIL.Image.crop()$ to isolate the extreme right edge — where $(\Delta m^2)$ reaches its maximum within the CL. Finally interpret scale and answer.

While these reasoning trajectories during RL exploration are not without flaws, e.g. occasionally exhibiting imprecise coordinate estimation or redundant tool calls, they collectively demonstrate the potential of tool-augmented multimodal reasoning.

Figure 12: Reasoning trajectories on tools composition.

## C    LIMITATIONS

While our method demonstrates promising emergent behaviors and strong performance across diverse visual reasoning tasks, several limitations remain. First, the reliance on high-quality synthetic trajectories implies that certain real-world reasoning patterns may be underrepresented, potentially limiting robustness in open-domain scenarios. Second, although code provides a universal interface, extending to richer modalities (e.g., audio) or domain-specific tools (e.g., medical applications) will require additional engineering. Finally, due to compute constraints, our evaluations are primarily conducted on 7B-scale models; the scalability of emergent behaviors at larger scales remains to be systematically examined. Nevertheless, our preliminary experiments suggest a promising trend when scaling up model capacity and compute resources.

## D    BROADER IMPACT

This work contributes toward building more transparent and verifiable multimodal reasoning systems by adopting executable code as the unified medium for tool use. The ability to generate interpretable traces and intermediate artifacts can benefit applications where accountability and auditability are essential, such as scientific analysis and education. At the same time, code-generating models pose risks: malicious users could potentially exploit them for unsafe automation, and generated visual artifacts might be misused to mislead or manipulate. To mitigate these concerns, we recommend pairing such systems with appropriate safeguards, including safety filters, usage constraints, and responsible deployment practices. By doing so, the benefits of executable visual reasoning can be realized while minimizing the potential for misuse.

## LLM USAGE STATEMENT

We used large language models (LLMs) only as auxiliary tools to correct occasional grammatical errors and typos throughout our writing process, and importantly, no parts of the paper were generated by LLMs in a substantive or large-scale manner. In addition, we confirm that LLMs did not contribute to research ideation, methodology design, training data synthesis and generation, or experimental analysis. We further emphasize that our submission contains no hidden prompt injections or any other misuse of LLMs.

