# OpenReview forum: "Executable Visual Reasoning: From Universal Solver to Emergent Behaviors"
_ICLR.cc/2026/Conference — ICLR 2026 Conference Withdrawn Submission_

### Official Review · Reviewer_k7Fr · 2025-10-22

**Soundness:** 3
**Presentation:** 3
**Contribution:** 3
**Rating:** 8
**Confidence:** 4

**Summary:**

The paper proposes ExeVision, a multimodal reasoning framework that treats executable code as a universal medium for visual reasoning. The system couples an MLLM with a sandboxed code executor to perform atomic visual–symbolic operations, interleaved with natural-language reasoning. The paper reports “emergent” behaviors during RL, i.e., novel code routines, tool compositions, and cross-task transfer, and shows improvements on some benchmarks versus text-only or schema-based baselines and some tool-using MLLMs.

**Strengths:**

The main idea, using executable code as a general reasoning substrate for images, is clear and timely. It offers a concrete path to verifiable, step-wise perception+reasoning beyond text-only CoT. The training recipe is sensible: SFT on atomic skills then RL with a two-level reward (sequence-level and turn-level) that explicitly controls tool usage.

The paper provides multi-bench evidence (counting/search/chart/math) with clear deltas vs. the 7B base (Qwen-2.5-VL-7B) and several open/closed models; Table 1 and Table 2 are helpful.

I appreciate the ablation on reward design showing RBAT vs outcome-only and a “reward every successful tool” variant, along with turn statistics and entropy trends; this addresses the common tool-spamming failure mode.

The dataset curation pipeline (weak-to-strong filtering, atomic supervision categories, template for code-augmented responses) is described with enough detail to be reproducible at a high level.

Overall, the paper is of good quality, and I quite enjoy reading it.

**Weaknesses:**

The following are my concerns/questions, which I expect to discuss with or be addressed by the authors.

1) I think the claims of “emergence” need stricter tests and quantification.
Currently, “novel tool attempting / transfer / composition” is primarily illustrated via trajectories and qualitative plots (see the emergent examples around the RL section). There is no operational metric of novelty nor a controlled test that rules out SFT leakage or prompt-level priors. Please introduce a held-out atomic-skill exclusion protocol where certain ops (e.g., grid overlay) are completely absent from SFT but appear during RL; measure frequency and success rate of those ops across seeds.
Also can add counterfactual ablations: disable the sandbox during RL but keep the same rewards to check whether “emergence” still shows up (it shouldn’t). Further suggestion: authors can provide quantitative counts of novel API calls per 1k rollouts and the downstream accuracy lift attributable to those calls. (This concern targets the “emergent behaviors” description near the examples and the RL overview.)

2) RBAT design might be missing sensitivity analysis and bounds.
Eq. (1) defines ($R_{\text{seq}}$) with decay ($d=\frac{1+\tanh(\gamma(0.5-\mu_{\text{acc}}))}{2}-\delta$), but the paper does not detail the effective range of (d), the choice/impact of ($\gamma$) and ($\delta$), or stability across seeds. Lines with Eq.(1)–(2) list fixed ($\gamma=4,\delta=0.2,\beta=0.2$) but no sweep. So,
   - Please report sweeps over ($\gamma,\delta,\beta$) and show accuracy vs. average turns vs. execution-failure rate.
   - Can the authors provide theoretical or empirical bounds preventing negative (d) or degenerate incentives?

3) I am concerned about the fairness of baseline comparisons and eval protocol.
Table 1/2 aggregate many external systems, but evaluation budgets (max turns, tool availability, resolution preprocess, OCR/detector access) can change results. You set max-turn=10 for eval, 6 for training (Implementation Details), but do the baselines get equivalent interaction budgets and high-res routing? For each baseline, it will be better to state turn budget, tool list, and image-resolution handling used during evaluation (ideally run them under your harness). Where you cite third-party numbers, mark them and provide a side-by-side re-run subset under your settings via VLMEvalKit; otherwise, discuss any remaining mismatches.

4) Data curation may need clearer provenance and leakage controls.
The 34k SFT set is assembled from public sources via weak-to-strong filtering; however, several leaderboards (e.g., ChartQA, Pixmo, HR-Bench) might overlap in image style or question templates with your synthetic prompts. It will be helpful to publish exact source lists and sampling filters; add a train–test overlap audit (hash-based for images and n-gram for question text). Also show per-benchmark results with and without near-duplicate filtering to verify robustness.

----

There are a few minor typos/formatting issues in the code snippets (e.g., mismatched quotes and line wraps in the PIL/OpenCV examples) that can be cleaned to avoid confusion; also avoid referring to non-included figures in the main text. See code blocks and captions in the example sections.

**Questions:**

1. Can you provide a quantitative emergence score (e.g., proportion of rollouts that call an operator unseen in SFT, success rate conditioned on that, and lift over a no-sandbox RL run)? Where exactly are the operator whitelists for SFT vs. RL logged?

2. How robust is RBAT to hyperparameters? Please share sweep plots for Table 1/2. Also confirm that the advantage mixing ($A_{\text{seq}}+A_{\text{turn}}$) is not sensitive to normalization choices.

3. What is the sandbox policy (time/memory, allowed imports, disabled syscalls)? Please include a short table with failure categories and rates.

---

### Official Review · Reviewer_REZ6 · 2025-10-30

**Soundness:** 3
**Presentation:** 2
**Contribution:** 2
**Rating:** 2
**Confidence:** 4

**Summary:**

The paper presents a code-executing multimodal agent that interleaves language “thinking” with Python tool calls. Training mixes SFT on executable trajectories with an RL stage using a “balanced” reward (RBAT) to curb tool spamming. The model is evaluated across counting, visual search, chart QA, and math-vision, with qualitative “emergent” behaviors highlighted.

**Strengths:**

1. Good analysis of emergent behaviors with concrete rollouts; the examples and insights are informative.
2. Reward ablations are useful; the paper digs into design choices behind the turn-level penalty vs. sequence-level accuracy.

**Weaknesses:**

1. **Positioning vs. prior program-execution agents is soft.** Compiling reasoning into executable code with iterative perception-action loops has clear precedent (e.g., ViperGPT and related program-execution pipelines). The interleaved “think-execute” loop and $R_{BAT}$ are incremental; the paper should make a crisper case for what is fundamentally new beyond (i) a broader tool set, (ii) the specific $R_{BAT}$ shaping ($R_{DeepEyes}$ vs $R_{BAT}$), and (iii) anecdotal emergence. Related work reads comprehensive but does not isolate the novelty gap.


2. **Result mismatches / provenance unclear.** Main Table 1 appears to quote baseline numbers that differ from the original papers. Concretely, for DeepEyes on HR-Bench-8K, I expect 72.6 (as reported in their paper), but your table shows a different value. Please reconcile.
 Further, for datasets like CountBenchQA, DeepEyes did not report results; the paper doesn’t explain how those entries were produced (re-runs with their open-source code? a re-implementation? different resolution/turn budgets?). If you re-ran open models, why are some other models/datasets omitted? The selection policy needs to be stated so readers can trust cross-paper comparisons.

**Questions:**

1. Table 3 notation. What are the small-font numbers underneath each score and what does their color encode? Are they average turn counts? If yes, how do we get 0 turns on datasets like CountBench?
2. RBAT sensitivity. Please report performance vs. the key RBAT coefficients and show at least one setting where over- or under-penalizing turns degrades accuracy, to justify “balanced.”

Minor Edits:
1. Equation (2), line 299: missing A_turn

---

### Official Review · Reviewer_7nM3 · 2025-10-31

**Soundness:** 2
**Presentation:** 3
**Contribution:** 2
**Rating:** 4
**Confidence:** 3

**Summary:**

This paper introduces ExeVision, a framework that performs visual reasoning by generating and executing code. Instead of relying solely on text-based reasoning, the model iteratively writes and runs code snippets to process visual inputs in a multi-turn loop. The authors also introduce a new reward design to balance tool exploration and efficiency to prevent overuse or underuse of tools. Experiments show that ExeVision outperforms schema-driven and texxt-only baselines.

**Strengths:**

S1. The idea of generating executable code for visual reasoning is intuitive and somehow interesting.

S2. The paper presents interesting findings from RL training, such as novel tool combination and cross-domain transfer.

S3. The paper is written well with examples to help understand the paper.

**Weaknesses:**

- As a counterpoint to S1, the idea of generating code for reasoning is not particularly novel. Prior works have already explored using code as an intermediate step for solving reasoning tasks (e.g., [1], [2]). Moreover, the concept is quite similar to tool calling—the only difference being that this paper attempts to synthesize the “tool” code dynamically. However, this design choice is debatable: generating code on the fly increases flexibility but also introduces potential synthesis errors, code injection risks, and higher token consumption compared to using existing tools. From this perspective, the contribution of synthesizing code for visual reasoning feels incremental rather than fundamentally novel, and the advantage of this design choice remains questionable.

- Another concern lies in the experimental evaluation. The authors compare ExeVision-7B with the Qwen2.5-VL-7B; however, the baselines use single-turn reasoning, while ExeVision-7B performs multi-turn reasoning with code generation. Therefore, it’s unclear whether the reported performance gains actually come from executable visual reasoning or simply from allowing more reasoning steps. A stronger baseline would be to compare the model with tool calling rather than synthesizing the tool code.

- DeepEyes appears to be a concurrent work cited in the paper. After checking the original DeepEyes paper, I found that the current work is largely an incremental extension of it. DeepEyes-7B uses a very similar approach, and the performance comparison between DeepEyes-7B and ExeVision-7B is a mixed bag -- sometimes DeepEyes-7B performs better, and in other cases ExeVision-7B is  better. Therefore, it’s unclear what the real advantage of synthesizing code is compared to directly using tools. A deeper analysis comparing the two models, clarifying under which conditions each performs better and why, would be very helpful. Moreover, the reported results seem to come from a single evaluation run without standard errors, making it difficult to tell whether the differences are due to just random variance.

- The proposed reward function includes multiple non-trivial components (e.g., gamma, delta) whose effects are insufficiently justified. The meaning and interpretation of d in Equation (1) are also unclear. Although an ablation study of the reward design is provided, it does not clarify how these parameters were chosen or how sensitive the model is to them. The reward design appears largely trial-and-error based, lacking clear theoretical motivation or intuitive explanation.

---

References:

[1] PAL: Program-aided Language Models (ICML 2023)

[2] Compositional visual reasoning without training (CVPR 2023)

**Questions:**

- Typo: In Table 2, Qwen2.5-VL-72B is categorized under closed-source MLLMs.

- How many reasoning turns are allowed during evaluation, and whether this setting is consistent across baselines such as DeepEyes?

---

### Official Review · Reviewer_Xf4a · 2025-11-01

**Soundness:** 2
**Presentation:** 3
**Contribution:** 2
**Rating:** 2
**Confidence:** 3

**Summary:**

In open-source multimodal large language models, chain-of-thought reasoning relies on text-only chains, while the visual input remains static. This paper introduces ExeVision, which utilizes code as chain-of-thought steps that manipulates images. The paper introduces a dataset of 34k COT trajectories using atomic code operations such as cropping and annotating. ExeVision first fine-tunes a MLLM on the curated dataset to improve model reasoning. It then is further improved using reinforcement learning through a novel reward function that balances code usage and accuracy. Results show that ExeVision outperforms base models significantly.

**Strengths:**

The paper is easy to follow with good visuals.

Code as a medium for CoT is effective.

The reward design balancing code usage and performance is simple and intuitive.

**Weaknesses:**

The overall method is not new. Fine-tuning LLMs using COT traces + RL have been shown to be effective [1].

The paper claims that COT is static and “prevents models from interacting with visual inputs or incorporating new observations during intermediate reasoning”. However, there is many existing work that utilizes multi-modal chain-of-thoughts which does not rely on static visuals [2, 3].

The title and various parts of the paper refers to code as “Universal Solver”. This is not clearly defined. Does the code being written in a general language like python make it a universal solver? Can code alone solve the visual reasoning problems?

The emergent abilities are not well-examined. The paper does not study how often these abilities occur, or if they are correctly used by the model.

[1] Guo, Daya, et al. "Deepseek-r1: Incentivizing reasoning capability in llms via reinforcement learning." arXiv preprint arXiv:2501.12948 (2025).

[2] Hu, Yushi, et al. "Visual sketchpad: Sketching as a visual chain of thought for multimodal language models." Advances in Neural Information Processing Systems 37 (2024): 139348-139379.

[3] Wu, Wenshan, et al. "Mind's eye of LLMs: visualization-of-thought elicits spatial reasoning in large language models." Advances in Neural Information Processing Systems 37 (2024): 90277-90317.

**Questions:**

Are there experiments that show how the emergent abilities are used by the model in a measurable way?

How does the proposed method differ from existing multi-modal chain-of-thought methods?

Were there other atomic abilities that you considered? How are the atomic abilities decided?

---

### Note · Authors · 2025-11-14

I have read and agree with the venue's withdrawal policy on behalf of myself and my co-authors.